# Development of a Good Clinical Practice inspection checklist to assess clinical trial sites in Vietnam

Quang Ngo Nguyen,[1] Nam Vinh Nguyen ![ORCID],[2,3] Dung Thi Phuong Nguyen,[2] Celine Vidaillac,[2] Phuong Thi Minh Trinh,[1] H Rogier van Doorn,[2,4] Guy E Thwaites,[2,4] Evelyne Kestelyn,[2,4] Ha Thi Nhi Vo[1]

QNN and NVN are joint first authors.
EK and HTNV are joint last authors.

¹Administration of Science Technology and Training, Ministry of Health of Vietnam, Ha Noi, Hanoi, Viet Nam
²Oxford University Clinical Research Unit, Ho Chi Minh city, Viet Nam
³Department of Pharmaceutical Management and Economics, Hanoi University of Pharmacy, Hanoi, Viet Nam
⁴Centre for Tropical Medicine and Global Health, Nuffield Department of Clinical Medicine, University of Oxford, Oxford, UK

**Correspondence to**
Nam Vinh Nguyen;
namnv@oucru.org

## ABSTRACT

**Background** Assessing the capacity of a healthcare institution to conduct and manage clinical research studies is challenging, especially in developing countries where resources are limited. The objective of this study was to develop a practical and transparent tool for the Vietnam Ministry of Health (MOH) to assess institutions' capacity to lead clinical trials in line with local and international regulations.

**Methods** We reviewed the literature, relevant official international and national guidelines, regulations and checklists for clinical sites' assessment to identify key indicators of clinical research capacity. We developed a Good Clinical Practice (GCP) inspection checklist consisting of a questionnaire with 30 key criteria, including 16 core criteria and 14 recommended criteria, related to four central aspects of clinical research management (ie, governance, operations, infrastructures and human resources). Following a detailed review and assessment by a panel of experts, sponsors and academic investigators, we assessed the checklist's applicability in a pilot study involving 10 sites with various clinical research experiences.

**Results** Independently of their clinical research experience, all participating institutions fulfilled most of the core criteria. In contrast, a significant variability was observed in the compliance to recommended capacity criteria, especially those related to governance (certifications and reporting) as well as operations (existence of a clinical research coordination unit or electronic trial management system).

**Conclusions** A GCP inspection checklist was successfully developed to support the MOH in the assessment of institutions' capacity to conduct clinical research. Additional efforts from all stakeholders are now warranted to provide local sites with sustainable capacity development resources that will further build up and harmonise Vietnamese clinical research settings.

## Strengths and limitations of this study

► This is the first Good Clinical Practice (GCP) inspection checklist developed for Vietnam to support the standardisation of clinical trial sites according to four key components of clinical research (governance, operations, infrastructures and human resources).
► The checklist was reviewed by a panel of experts including representatives from Vietnam regulatory agency, trial sites, international sponsors, contract research organisations and academic institutions to ensure compliance to local and international regulations.
► The applicability of our inspection checklist was assessed through a pilot study involving 10 sites with various clinical research experiences located across Vietnam.
► Purposeful sampling of the studied sites for the pilot study might result in a selection bias, since sites willing to participate to the study could have felt confident about their compliance to GCP standards.
► Among the invited sites to our pilot study, only one refused to participate, suggesting that the participation bias was negligible.

and scientifically sound conduct of clinical research.[1] Guidance and regulations on the different aspects of conducting clinical research are available from several national and international institutions, such as the Council for International Organizations of Medical Sciences (CIOMS), the European Agency for the Evaluation of Medicinal Products and the US Food and Drug Administration (FDA).[2–4] In addition, most countries have adopted the International Conference on Harmonization Good Clinical Practice (ICH-GCP) guidelines or formulated country-specific guidelines based on the original ICH-GCP framework.[5] Some countries have also established structures and processes to evaluate research institutions' compliance to applicable guidelines, as well as their ability to conduct trials appropriately.

## INTRODUCTION

Whether a clinical trial is conducted in an economically developed or resource-limited country, adherence to local and international standards and principles of Good Clinical Practice (GCP) is generally recognised as an important requirement for the ethical

In resource-limited countries, there is often a tension between compliance to international standards with related costs for implementation and oversight, and the need to be pragmatic and adapt to a given context with existing structural, cultural and economic challenges.[6 7]

With nearly 200 hospitals and an approximately 180 active research studies (which include clinical trials, patient registries and observational studies), Vietnam is emerging as an important global participant in the clinical trial landscape (ClinicalTrials.gov). Due to the increasing complexity of clinical trials and the growing size of populations enrolled, the Vietnamese authorities have recognised the need to closely monitor the context and environment in which clinical research studies are conducted. Concerns have been raised nationally about the capacity of local sites to conduct clinical trials while complying with international standards and national regulations.[8] The sites with the highest potential for patient recruitment are often the busiest hospitals, where human, technical and financial resources are mainly allocated for patient care, rather than research activities.[1 9–11] To respond to these growing concerns, the Vietnamese authorities have recently mandated the Administration of Science Technology and Training, within the Ministry of Health (MOH-ASTT) and the Oxford University Clinical Research Unit to develop a standardised, transparent and quantitative GCP inspection checklist to facilitate and support the assessment of the clinical trial capacity of Vietnamese hospitals.

The aim of the checklist is to assess the existence of efficient governance and operational processes, adapted infrastructures and qualified human resources, in order to fulfil all the legal and medical requirements set by the national regulatory authorities, sponsors and international GCP guidelines.[1 11 12] The checklist will be used by the MOH to assess a new site's capacity to start conducting clinical research and for its regular site audits to ensure sustained quality and good practice across all clinical research sites in Vietnam. Additionally, the generated data will provide an up-to-date overview of the research capacity in Vietnam and can inform international sponsors and clinical research organisations about the capacity of any Vietnamese institution to conduct clinical research. This manuscript describes the development and review of the GCP inspection checklist as well as its first application in a pilot study involving 10 healthcare institutions in Vietnam.

## METHODS
### Checklist development process
The checklist development was initiated by reviewing Vietnamese legal documents and current national guidelines for clinical trial conduct. Those documents include: (1) Circular 29/2018/TT-BYT on the Vietnamese regulations on clinical trials on drugs and (2) Circular 45/2018/TT-BYT on the regulations on the establishment, functions, tasks and rights of the ethical committees (ECs) in biomedical research.[13 14] In order to identify key indicators of ethical and scientifically

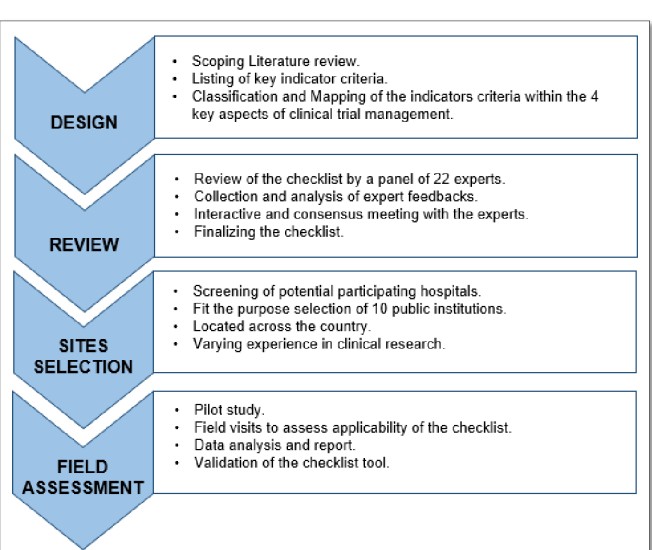

**Figure 1** Process of development of a Good Clinical Practice inspection checklist to assess sites' capacity to conduct clinical research in Vietnam.

sound clinical trial management and ensure harmonisation with international standards in clinical trials, the MOH-ASTT team (QNN, NVN and HTNV) also reviewed guidelines released by international health agencies (eg, FDA, EMA, CIOMS and ICH-GCP) and public/non-profit research institutions (eg, Oxford University Clinical Research Unit, Duke Global Health Institute).[2–4 15 16] The team selected 30 key criteria from the reviewed literature as a framework for the initial GCP inspection checklist. A key criteria was selected if it was either related to aspects regulated by the current Vietnamese laws (core criteria) or not yet regulated by the Vietnamese laws but mentioned in at least two guidelines of international health agencies and research institutions (recommended criteria). Those criteria were classified into one of four key aspects of clinical trial management: governance, operations, infrastructures and human resources (figure 1).

Current Vietnamese health regulations were revised in parallel with the literature review to ensure the applicability of the initial checklist within the overall legal context related to healthcare services and science and technology in Vietnam (table 1). The initial checklist was then sent via email for review to a panel of 22 local experts from institutions that had collaborated with the MOH-ASTT to develop legal framework for clinical trial management (table 2). Individual experts feedback was elicited, analysed and synthesised. The experts were next invited to a workshop organised by the MOH-ASTT in Hanoi on the 14 April 2017, where the team presented their feedback, listed the criteria with consensus feedback between experts and highlighted those for which there were conflicting reviews. All the invited experts participated and were asked to interactively discuss these 30 criteria and provide further insights. The main objective of this workshop was to achieve an agreement by the panel of experts and future users on all the criteria of

**Table 1** List of laws and regulations reviewed to develop the Good Clinical Practice inspection checklist

| Document title | Vietnamese title | Access link | Note |
|---|---|---|---|
| **Legal documents in support of developing the checklist** | | | |
| Decision N\no. 779/QĐ-BYT on releasing guideline on 'Good Clinical Practice' | Quyết định số 779/QĐ-BYT về việc ban hành 'Hướng dẫn thực hành tốt thử thuốc trên lâm sàng' | http://asttmoh.vn/document_cat/van-ban-phap-quy-2/ | The development of the checklist was not fully compliant with these documents since they were released many years ago. It was also based on the draft version of Circular no. 29/2018/TT-BYT – Regulations for Clinical Trials on Drugs, which was made in December 2016. Vietnamese |
| Circular no. 03/2012/TT-BYT on guiding clinical trials on medications | Thông tư 03/2012/TT-BYT Hướng dẫn về Thử thuốc trên lâm sàng | http://vbpl.vn/boyte/Pages/vbpq-van-ban-goc.aspx?ItemID=27303 | |
| Official dispactch no. 6586/BYT-K2ĐT on defining and reporting of SAEs in clinical trials | Công văn 6586/BYT-K2ĐT Hướng dẫn báo cáo, ghi nhận SAE trong TNLS | http://asttmoh.vn/wp-content/uploads/2014/12/Bao-cao-SAE.pdf | Similar to above, the development of the checklist was also based on the draft version of Decision No. 62/QĐ-K2ĐT on guiding detecting, managing and reporting AE, SAE in clinical trials, which was made in Octorber 2016. Vietnamese |
| **Legal documents reviewed after the development of the checklist** | | | |
| Law no. 40/2009/QH12 on medical examination and treatment | Luật Khám chữa bệnh 2009 (Luật số 40/2009/QH12) | https://kcb.vn/vanban/luat-kham-benh-chua-benh-2 | Vietnamese |
| Law no. 105/2016/QH13 on pharmacy | Luật Dược 2016 (Luật số 105/2016/QH13) | http://extwprlegs1.fao.org/docs/pdf/vie178850.pdf | English (unofficial) |
| Decision no. 111/QĐ-BYT on the organisation and operation of local IRB | Quyết định số 111/QĐ-BYT về việc ban hành Quy chế Tổ chức và hoạt đồng của Hội đồng đạo đức trong nghiên cứu y sinh học cấp cơ sở | http://asttmoh.vn/wp-content/uploads/2014/12/Q-111-v_-quy-ch_-ho_t-_ng-c_a-IRB.pdf | Vietnamese |

IRB, Institutional Review Board.

the checklist prior to be tested in a pilot study involving 10 institutions across Vietnam.

### A pilot study to assess the applicability of the capacity indicator checklist

After adapting the checklist based on the experts' feedbacks, a pilot study including 10 institutions with previous experience in leading or participating in clinical trials was conducted. Participating sites were purposefully selected to include public hospitals with different specialisations (general medicine, tropical diseases and vaccination, haematology paediatric, surgery, geriatrics and obstetrics/gynaecology), including a variety of locations, activities (specialised vs general hospitals) and experience in clinical trials (leading vs non-leading sites). The checklist was sent to the hospital 2 days prior the site visit by a field assessment team that included two specialists from the MOH-ASTT and one non-governmental expert. On the day of the visit, the team inspected the site and conducted interviews to assess each criteria of the checklist. Thus, fulfilment of indicators was defined by the answers from

the staff and by the evidence provided. For criteria related to infrastructure, field visits were conducted, and fulfilment of the related criteria was defined by the combination of three elements: (1) existence of suitable infrastructure, (2) existence of standardised documents and (3) any other relevant documents attesting the use of the facilities during clinical trials such as equipment and accountability logs, etc. The field visits took place at the 10 selected institutions, from 1 to 26 October 2017. This procedure was designed to simulate an external audit to validate the toolkit and ensure its suitability.

### Patient and public involvement

Patients were not involved both in the design of the checklist and the implementation of our pilot survey.

## RESULTS

### Expert review of the GCP inspection checklist

The final checklist included 30 criteria covering four key aspects of clinical trial management: (1) governance, (2)

**Table 2** Description of the panel of experts involved in the design of the Good Clinical Practice inspection checklist

| Characteristics | Number (%) |
|---|---|
| Gender | |
| Male | 16 (72) |
| Female | 6 (28) |
| Age | |
| <30 | 2 (9) |
| 30–50 | 11 (50) |
| >50 | 9 (41) |
| Institution | |
| ASTT-MOH | 1 (4.5) |
| Drug Administration of Viet Nam | 1 (4.5) |
| Medical Service Administration | 1 (4.5) |
| IEC-MOH | 2 (9) |
| Clinical trial sites | |
| Bioethics committee members | 6 (27) |
| Principal investigators | 2 (9) |
| Research managers | 2 (9) |
| Clinical trial service providers | 2 (9) |
| Sponsors | 2 (9) |
| Local academic institutions | 2 (9) |
| National pharmacovigilance centre | 1 (4.5) |

ASTT-MOH, Ministry of Health – Administration of Science Technology and Training; IEC-MOH, Ministry of Health – Independent Ethic Committee.

operations, (3) infrastructures and (4) human resources. Indicators currently regulated by Vietnamese laws and regulations (16/30) were classified as core or mandatory criteria, while aspects (14/30) not yet regulated by the Vietnamese laws were classified as recommended criteria (figure 2)

Core governance criteria included: (1) knowledge (GC01) and up-to-date standards (GC03) of Good Healthcare Practices, which cover ICH-GCP and Vietnamese GCP, Good Storage Practices and Good Laboratory Practices[13 17 18]; (2) existence of internal regulatory mechanisms related to clinical trials management (GC02), (3) existence of a systematic approach to manage and store essential documents related to clinical trials (GC04); and (4) existence of a local bioethics committee (GC05). Recommended governance criteria included: (1) existence of a valid business registration certificate (GR01)[19] and a valid science and technology enterprise certificate (GR02),[20] which are both granted by the Vietnamese government, (2) existence of internal guidelines or standard operating procedures (SOP) related to clinical trial management (GR03), (3) existence of an International Organization for Standardization (ISO) 9001:2015 certificate for Clinical Trial Supporting Activities certification

or equivalent (GR04); and (4) clean record of safety reporting for completed or ongoing trials (GR05).

Core operational criteria included: (1) existence of an internal scientific and training unit (OC01) and (2) regular (once a year) examination by the study team of drug storage facilities (OC02) and equipment used for routine diagnosis in clinical trials (OC03). Recommended operational criteria included: (1) existence of an independent clinical trial unit (OR01), (2) regular (>1×/month) EC meeting (OR02)) (3) existence of an electronic clinical trial management system (OR03); and (4) centralised storage and management of study drugs by qualified professionals (pharmacy unit) (OR04).

Core infrastructure indicators covered administrative facilities (IC01), general medical facilities (IC02), emergency care (IC03), pharmacy, (IC04) and diagnosis equipment available (IC05).

Lastly, core human resources indicators included: (1) regular GCP training for investigators (HC01) and clinical research managers (HC03) and (2) regular training of investigators on safety reporting and management of clinical trials (HC02). Trainings must be renewed every 3 years as per current Vietnamese regulation. Recommended human resources criteria included: (1) GCP training for hospitals managers (HR01), clinical research nurses (HR02) and pharmacists (HR03) and (2) training of investigators on methodology (HR04) and risk management (HR05) of clinical trials.

Experts were overall satisfied by the checklist through the feedback that we collected as well as their interactive discussion in the workshop but emphasised the importance of the investigators' training on safety reporting and GCP training of research managers to ensure engagement of the PI and study team towards a better compliance to current national and international GCP standards. Although the current Vietnamese laws do not regulate the recommended criteria covered in the checklist, experts felt compliance to these criteria should be closely monitored, as it would reveal active internal mechanisms deployed by local institutions to facilitate the implementation and the conduct of clinical trials per local and international standard guidelines (figure 2). Therefore, these two criteria, which were initially recommended criteria in the first draft version of the checklist developed by the MOH-ASTT team, were recategorised into core criteria in the final checklist.

### Pilot study of the GCP inspection checklist

The pilot study included five clinical trial sites in Hanoi, one in Hue and four in Ho Chi Minh city. Experience in clinical trial implementation and management of the participating hospitals varied widely, with half of the sites having minimal practice in leading clinical trials. Results of the pilot study showed that the most experienced institutions (leading sites with >10 clinical trials in their pipeline) fulfilled all the core criteria except for one criteria relating to the training of investigators on safety reporting and management in clinical trials (figure 3A).

| GOVERNANCE | | OPERATIONS | | INFRASTRUCTURES | | HUMAN RESOURCES | |
|---|---|---|---|---|---|---|---|
| CORE INDICATORS | | | | | | | |
| GC01 | GxP regulations to implement clinical trials are satisfied. | OC01 | The site has an internal scientific unit to implement clinical trials. | IC01 | The site has suitable administrative facilities for the management of clinical trials. | HC01 | All study Investigators are regularly trained on GCP. |
| GC02 | Internal policies to manage clinical trials are in place at the site. | OC02 | Regular examination by the study team of study treatments storage facilities are performed. | IC02 | The site has suitable facilities for the general medical requirements of clinical trials. | HC02 | All study Investigators are regularly trained on safety reporting and management of clinical trials. |
| GC03 | GxP regulations related to clinical trials are up to date and documented. | OC03 | Regular examination by the study team of diagnosis equipment used in clinical trials are performed. | IC03 | The site has suitable facilities for emergency care if required in clinical trials. | HC03 | Clinical research managers are regularly trained on GCP. |
| GC04 | Existence of a systematic approach for management and storage of essential documents. | | | IC04 | The site has suitable facilities for the storage of study treatments. | | |
| GC05 | Site has a bioethics committee in place. | | | IC05 | The site has suitable diagnosis equipment for clinical trials. | | |
| RECOMMENDED INDICATORS | | | | | | | |
| GR01 | The site has valid business certificates (ERC) granted by the government. | OR01 | The site has a clinical trial coordination unit. | | | HR01 | Hospital managers are trained on GCP. |
| GR02 | The site has a valid certificate of science and technology enterprise granted by the government. | OR02 | The bioethics committee of the site meets at least 1 per month. | | | HR02 | Clinical research nurses are trained on GCP. |
| GR03 | The site has internal guidelines and/or SOPs related to clinical trial management. | OR03 | The site has an electronic management system for trial documents. | | | HR03 | Study pharmacists are trained on GCP. |
| GR04 | The site has ISO (or similar standards) to attest of the quality of management systems in place for ongoing trials. | OR04 | The site has a qualified professional to manage the storage of study treatments (e.g. Pharmacy unit). | | | HR04 | Study investigators are trained on methodology in clinical trial. |
| GR05 | The site is up to date for SAE reporting of completed or ongoing trials. | | | | | HR05 | Study investigators are trained on risk management of implementing clinical trials. |

**Figure 2** Good Clinical Practice inspection checklist developed to assess the clinical research capacity of healthcare institutions in Vietnam. ERC, enterprise registration certificate; GC, core governance criteria; GCP, Good Clinical Practices; GR, recommended governance criteria; GxP, Good Healthcare Practices (GCP, GLP and GSP); HC, core human resources criteria; HR, recommended human resources criteria; IC, core Infrastructures criteria; IR, recommended infrastructure criteria; ISO, International Organization for Standardization; OC, core operational criteria; OR, recommended operational criteria; SAE, serious adverse event; SOP, standard operating procedures.

This criterion was fulfilled by 60% of sites, irrespective of the experience in clinical trials (figure 3B). Institutions with less clinical research practice (with <10 clinical trials in their pipeline) performed well with up to 11/16 core criteria fulfilled. Of note, 80% of the institutions with less experience in clinical research had a structure to support the implementation of clinical trials (criterion OC01), up-to-date internal policies for clinical trials management (criterion GC02), documented regulations on clinical trials (criterion GC03) and GCP trained clinical research managers (criterion HC03) (figure 3B), suggesting the efforts and dynamism deployed by local institutions to implement clinical research standards

All the leading institutions and 80% of the non-leading sites fulfilled the recommended criteria on human resources (HR01–HR04), highlighting an active and positive engagement of all the sites towards training of the staff involved in clinical research. However, significant variability was observed across all the sites for the recommended operational and legal aspects (figure 3C). Only 60% and 40% of the leading and non-leading sites, respectively, had a certificate to implement business activities (criterion GR01), and 80% and 60% of the leading and non-leading sites, respectively, had a certificate to implement scientific and technology activities (criterion GR02) (figure 3C). Only one site (10%) of the 10 participating institutions had a clinical trial coordination unit

(criterion OR01). Although all the sites evaluated had an established bioethics committee (criterion GC05), only one of the least experienced sites (20%) and four of the most experienced institutions (80%) had their bioethics committee meetings at least once a month (criterion OR02). Finally, none of the sites, regardless of their experience in clinical research, met the criteria for electronic management of clinical trial documents (criterion OR03), internal guidelines/standard operations procedures, ISO certificates and up-to-date reports for serious adverse events (criteria GR03-GR05) (figure 3C).

## DISCUSSION

Vietnam has been actively participating in international clinical trials over the past decades.[9] Hence most of the sites assessed in this study demonstrated a very good compliance to governance, operations, infrastructures and human resources core criteria. Our pilot study found that training of trial investigators on adverse event (AE) monitoring and reporting was the core criteria with the lowest score in both leading and non-leading hospitals. Serious adverse events (SAEs) and AEs are often under detected and under-reported.[21] Although our study was not designed to identify the causes for the poor compliance to this criterion, several reasons may be put forward. First, we found that clinical trial investigators generally

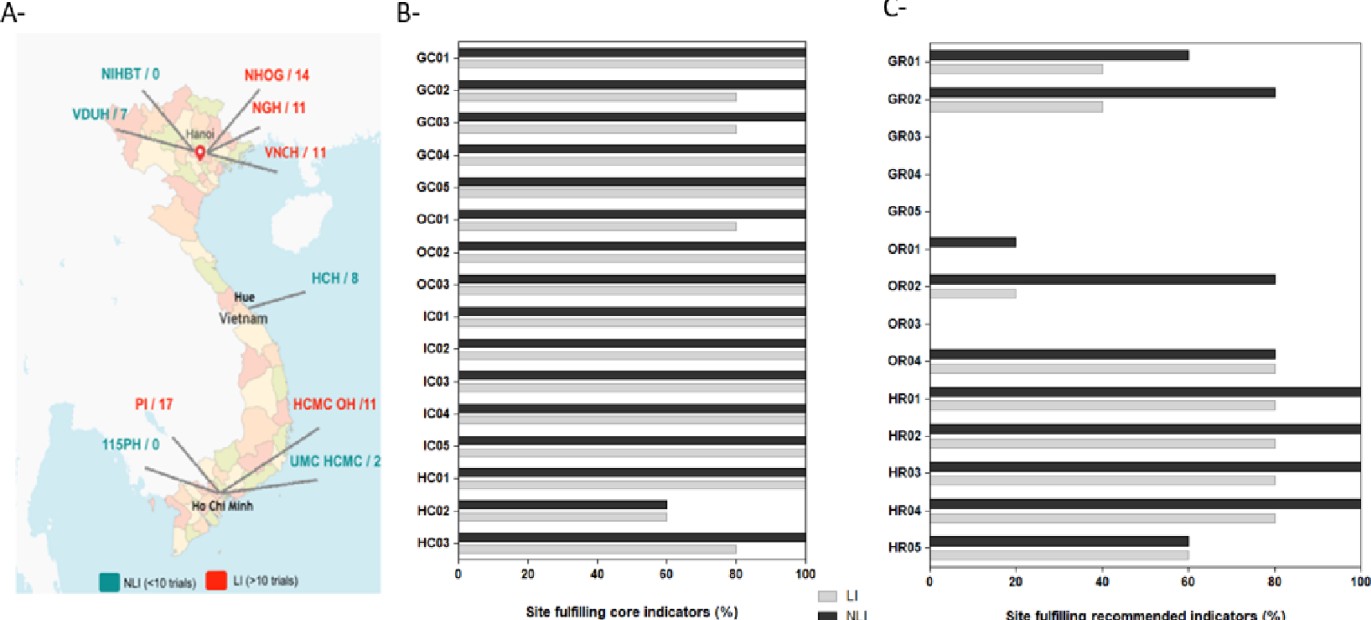

**Figure 3** Participating healthcare institutions and quantitative results of our checklist applicability pilot study. (A) Ten sites purposefully selected participated in the pilot study. Five institutions had little to no experience in the design, implementation and leadership of clinical trials (non-leading sites (NLIs)), while five other sites had already submitted protocols, implemented and led more than 10 clinical trials (leading institutions (LIs)). The map was downloaded from freevectormap (https://freevectormaps.com) and modified for publication purposes. (B) All the core governance (GC), operational (OC), infrastructure (IC) and human resources (HC) indicators were fulfilled by the LI with the exception one criteria (HC02). NLI met most of the core criteria (11/16), which demonstrates the potential of these sites to quickly adapt and offer an adequate environment for the conduct of clinical trials per national and international standards. (C) Variability in the recommended operational and governance indicators was found across all the sites. 115 PH, 115 People's Hospital; GC, core governance criteria; GR, recommended governance criteria; HC, core human resources criteria; HR, recommended human resources criteria; HCH, Hue Central Hospital; HCMC OH, Ho Chi Minh City Oncology Hospital; IC, core infrastructures criteria; IR, recommended infrastructure criteria, NGH, National Geriatric Hospital; NHOG, National Hospital of Obstetrics and Gynaecology; NIHBT, National Institute of Haematology and Blood Transfusion; OC, core operational criteria; OR, recommended operational criteria; PI, Pasteur Institute; UMC HCMC: University Medical Center Ho Chi Minh City; VDUH, Viet Duc University Hospital; VNCH, Viet Nam National Children's Hospital.

lacked knowledge and practice in terms of SAE identification, monitoring and reporting to ethic committees, regulatory agencies and the medical community in general. This represents an alarming issue since poorly detected and reported AEs/SAEs has proven to have an impact on the conduct of a trial, on its result's interpretation and ultimately on the future usage of the trial treatment in practice.[17] Second, trial investigators typically deal with high volume of clinical work not related to research or the trials they may be involved in. This affects their availability with the study participants to discuss, explore and identify AE-related symptoms that may have occurred between study visits or they may experience at the time of the visit. Thirdly, it has been shown that there is a discrepancy between patients and clinicians perspective of adverse events, with patients having a better appreciation of their underlying health status.[22 23] Thus, clinician appreciation of an adverse event may be influenced and sometimes biased by their own understanding of the severity and context of the disease or their confidence in the trial treatment to help the patients. This may ultimately lead the clinicians to downgrade AE-related symptoms and justify continuation of study treatment.[24 25] AE

reporting might also be influenced by sponsors when trial investigators implicitly transfer a part or all of their safety reporting duty to the contract research organisation hired by the sponsor.[21] All these important issues were highlighted and debated during our workshop, and led to a change in the classification of this criterion in the checklist and the incorporation of this training into the Vietnamese health regulations.[13]

Review of the recommended criteria highlighted the limitations that need to be addressed in order to improve clinical research site capacity in Vietnam. At the time of inspection, none of the institutions assessed had SOPs or ISO certifications in place to attest for the management and quality of their operational and governance systems. SOPs are not only critical to ensure standardised and qualitative procedures in line with the approved trial protocols, they also provide employees with detailed and clear roles and responsibilities, and support and sustain transparent work processes. This criterion received significant positive feedback from the panel of experts reviewing the checklist as it is believed to help institutions in the conduct of clinical trials according to regulatory approvals and institutional policies, contract agreements

with sponsors as well as national and international guidelines. Developing and implementing SOPs therefore contributes to a safer and better-managed work environment with consistent, high-quality output, which in turn makes the research institutions more attractive to study sponsors.[26] In addition, nearly half of the participating institutions lacked certificates to implement pharmaceutical and medical businesses. In recent years, the expansion of decentralised management and mechanisms for financial autonomy in Vietnamese public hospitals has required research institutions to frame research as an advanced clinical practice and as a business.

Our pilot study also showed a lack of available funding and expertise to manage clinical trials. Almost none of the visited sites had an independent trial management unit. In other countries, these multidisciplinary units have proven to be key in providing specific and expert services to support management and quality throughout all stages of a research project.[1] These include, for example, adapting research protocol to hospital settings, saving costs at implementation, collecting and managing high-quality data, improving patient recruitment patterns and strengthening AE/SAE monitoring and reporting.[12 26] In addition, although all of the hospitals assessed in this study had established local ECs to review the application and implementation of the clinical trials, their role and responsibilities remained limited and unclear. In Vietnam, ethical oversight responsibility for clinical trials has been shared between local ECs within hospitals and the National Independent Ethical Committee of the Ministry of Health (IEC-MOH) since 2013.[27] Local ECs have been established to enable the IEC-MOH to focus on multicentre national and international studies. The local ECs are responsible for evaluating the feasibility of a trial and its potential risks for study participants and reviewing/monitoring safety reports submitted by trial investigators. This high volume of workload requires timely and efficient collaboration between the members of these committees.[8 27] However, our pilot study show that only half of research hospitals had their ethics committee meet once a month, and core members of this committee were often key members of hospitals. Therefore, actual time spent by committee member on ethical matters is very limited. This weakens their roles and responsibilities in protecting the right and safety of volunteering patients and ensuring transparency and reliability of trial implementation.

Our study has a few limitations. First, most of the criteria assessed focus on the existence of mechanisms crucial for the conduct of clinical trials but do not evaluate the site's performance of the execution of these mechanisms. Thus, for example, obtaining a GCP certificate will not reflect the daily good clinical practice of the GCP trained investigator. Monitoring and audits will remain mandatory mechanisms to ensure the system in place complies with all relevant regulations and policies. Although we do acknowledge this limitation, the purpose of this checklist was for the MOH-ASST to assess the existence of key

mechanisms required to achieve the highest standards in clinical research in the context of Vietnam and not to assess the performance of the sites in their research activities. Besides, our pilot study was conducted in a small number of sites based on voluntary participation. This purposeful sampling of the studied sites for the pilot study might result in a selection bias since sites willing to participate to the study could have felt confident about their compliance to GCP standards. Finally, since only one invited site refused to participate, we believe the participation bias, if exists, remains minimal.

Building or strengthening research capacity has been increasingly recognised as one of the prerequisites to address health challenges and inform policy decisions in resource-limited countries.[7] Despite collective interventions through national and international programmes aiming at developing self-sufficient trial capacity, sites ability to lead clinical trials focused on improving local health needs, remain limited.[28] Healthcare professionals usually have little to no time for research and the operational challenges and workload that come with it. Compliance to international GCP standards therefore represents an enormous challenge for local institutions and investigators, due to lack of time and resources during the study.[12] In this context, we believe that the adoption of a GCP inspection checklist in support of assessing clinical trial site is crucial for Vietnam, as well as other resource-limited countries, which are emerging in clinical trials. Thus, our GCP inspection checklist can serve as reference for resource-limited countries looking at strengthening clinical research capacity.

## CONCLUSION

This study summarises the outcomes of a successful collaborative project between the Vietnamese Ministry of Health, policy makers and local investigators. The developed checklist is expected to further evolve as Vietnam will grow in the clinical research landscape and learns from its own experience. Of interest, this work has highlighted the lack of nationally available information and scientific reviews on institutions' clinical research capacity. Additional efforts from all stakeholders (sponsors, regulators and investigators) are now warranted to provide local sites with sustainable capacity development resources that will further build up and harmonise Vietnam clinical research settings. More discussions related to research accreditation in developed and resources-limited countries would greatly benefit the clinical trials community and guide countries that are emerging in the clinical research landscape. In a continuous effort to improve the quality of the research conducted in Vietnam, this study has inspired new collaborative projects such as the implementation of a tool to assess and remodel the operational characteristics of local research ethics committees or the development of new policies regulating drug safety reporting in clinical trials.

**Contributors** QNN: conceptualisation, investigation, writing – original draft, writing – reviewing and editing; NVN: conceptualisation, investigation, methodology, project administration, writing – original draft, writing – reviewing and editing; DTPN: conceptualisation, project administration, funding acquisition and writing – reviewing and editing; CV: methodology, visualisation, writing – original draft, writing – reviewing and editing; PTMT: project administration and investigation; HRvD: writing – reviewing and editing; GET: writing – reviewing and editing; EK: methodology, supervision, writing – original draft and writing – reviewing and editing; HTNV: conceptualisation, methodology, project administration, supervision and writing – reviewing and editing.

**Funding** This study was supported by funding from the Welcome Trust (106680/B/14/Z).

**Map disclaimer** The inclusion of any map (including the depiction of any boundaries therein), or of any geographic or locational reference, does not imply the expression of any opinion whatsoever on the part of BMJ concerning the legal status of any country, territory, jurisdiction or area or of its authorities. Any such expression remains solely that of the relevant source and is not endorsed by BMJ. Maps are provided without any warranty of any kind, either express or implied.

**Competing interests** None declared.

**Patient and public involvement** Patients and/or the public were not involved in the design, or conduct, or reporting, or dissemination plans of this research.

**Patient consent for publication** Not required.

**Ethics approval** The study does not involve human participants. It is purely a health system research and therefore does not require ethics committee's or institutional board's approval according to the regulation of Vietnam.

**Provenance and peer review** Not commissioned; externally peer reviewed.

**Data availability statement** Data are available on reasonable request. Data on pilot interviews can be approached on reasonable request. Please email to the corresponding author (namnv@oucru.org) for further information.

**ORCID iD**
Nam Vinh Nguyen http://orcid.org/0000-0002-9816-3371

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
