## [Reviewer comments · BMJ Open]

ARTICLE DETAILS

TITLE (PROVISIONAL)	DEVELOPMENT OF A GOOD CLINICAL PRACTICE INSPECTION CHECKLIST TO ASSESS CLINICAL TRIAL SITES IN VIET NAM
AUTHORS	Nguyen, Quang; Nguyen, Nam; Nguyen, Dung; Vidailac, Celine; Trinh, Phuong; van Doorn, H. Rogier; Thwaites, Guy; Kestelyn, Evelyne; Vo, Ha

VERSION 1 – REVIEW

REVIEWER	Hurtado-Chong, Anahi AO Foundation Office Dubendorf, AO ITC
REVIEW RETURNED	27-Feb-2021

GENERAL COMMENTS	The authors have done a great job in explaining their objective, methods and results. Starting with a review of the local legislation and international research guidelines/standards, they identified key items to elaborate a checklist that would enable a standardized assessment of clinical research sites in Vietnam. Then using a multidisciplinary panel to refine the key items, and finally testing it on a pilot study including sites with different levels. Discussion of key results and limitations is adequate and balanced. I congratulate the authors for this paper and recommend only minor revisions.
---

REVIEWER	Conroy, Tiffany Flinders University, College of Nursing and Health Sciences
REVIEW RETURNED	01-Apr-2021

GENERAL COMMENTS	Thank you for the opportunity to review your well written article. Your paper has the potential to be a useful roadmap for other resource-limited countries who are exploring similar issues. I offer the following suggestions with this in mind. It could be helpful to emphasise how your project could be replicated by other countries. While it is interesting to know the specifics for Viet Nam, the transferability of the information is not explicit. For example, you have prioritised as 'core' the Vietnamese legal document and guidelines. An explanation of why this decision was made and the alternatives that could be considered would be helpful and would knowing about the potential contrast and overlap in information contained in these documents with the international literature. Abstract: line 17 reference to core criteria and recommended capacity criteria is unclear as these are not explained until later in the paper. could limited to the 4 key aspects and introduce the other issues in the text. Line 91: It would be helpful to describe the 'team'. Was it all the authors? How were the key criteria 'selected'? What were the selection criteria? Were any criteria excluded?
---

	Lines 101-102: how were the experts selected, and what were their disciplines as this is not reported in Table 2. Did all invited experts respond and attend the workshop? Line 105: Which criteria were being discussed? All 30? Line 111: How were any criteria modified, added or removed after the workshop? What was adapted and why? Line 138:: refers to Table 2, but this seems to not fit here Lines 140-168: Repeats what is in Figure 2. The reason for having this detail here is not clear. Lines 170-173: How was expert satisfaction determined? On the advice of experts and you suggesting there is a hierarchy in the criteria, that is, is investigator training more important than some other criteria? Lines 247-250: The rationale for requiring a certificate to implement business activities is explained in the discussion but not before it is presented as a criteria. It may help to explain this prior. The discussion clearly explores the issues relating to recognition and reporting of adverse effects, and lack of SOP's and resources. Again, this is vital information for other resource-limited countries and the contribution of your work towards similar projects in these countries could be emphasised here. Figure 1 is not referred to in the text.
--	--

VERSION 1 – AUTHOR RESPONSE

Reviewer: 1

Dr. Anahi Hurtado-Chong, AO Foundation Office Dubendorf Comments to the Author:

Line 35: Revise to show that sites were 'purposefully selected'. A selection bias in the sites is possible in regard to their eagerness to participate in the pilot study.

=> We revised according to your comment: "Purposeful sampling of the studied sites for the pilot study might result in a selection bias, since sites willing to participate to the study could have felt confident about their compliance to GCP standards".

Table 1: Date of the draft

=> We add the date of draft as your comment

Table 2: "This seems to be a combination of institutions and positions/roles (eg ASTT-MOH, National Pharmacovigilance center vs principal investigators, research managers. I'd suggest to separate the two of them"

=> We revised this table according to your comment, it will highlight both institution (adding "clinical trial sites") and members included. We hope that you agree that it would be a good way to deal with the problem that you accurately highlighted

2. Reviewer: 2

Dr. Tiffany Conroy, Flinders University Comments to the Author:

Abstract: line 17 reference to core criteria and recommended capacity criteria is unclear as these are not explained until later in the paper. could limited to the 4 key aspects and introduce the other issues in the text.

=> We agree with your comment and add "including 16 core criteria and 14 recommended criteria" the abstract section

Line 91: It would be helpful to describe the 'team'. Was it all the authors?

=> We agree with your comment and added the names of the team. It was not all the authors, only the ones from the Administration of Science and Technology, the Vietnamese Ministry of Health

How were the key criteria 'selected? What were the selection criteria?

=> We agree with your comment and added "A key criteria was selected if it was either related to aspects regulated by the current Vietnamese laws (core criteria), or not yet regulated by the Vietnamese laws but mentioned in at least two guidelines of international health agencies and research institutions (recommended criteria)". We also slightly modified the text following it to avoid duplicate.

Were any criteria excluded?

=> There was no criteria excluded

Lines 101-102: How were the experts selected?

=> They were the experts who have collaborated with MOH-ASTT to develop previous legal documents related to clinical trial management. We included this in the text according to the reviewer's comment ("The initial checklist was then sent via email for review to a panel of 22 local experts from the institutions which had collaborated with the MOH-ASTT to develop legal framework for clinical trial management")

And what were their disciplines as this is not reported in Table 2.

=> We replaced "disciplines" by "institutions" in the text to match with the information in the Table 2.

Did all invited experts respond and attend the workshop?

=> All of them responded and attended to the workshop. We have edited lines 120-123 (marked copy) to reflect this.

Line 105: Which criteria were being discussed? All 30?

=> We added "30" to clarify. We did review all of 30 criteria again since we wanted a confirmation from all experts again in case they might have skipped the criteria through their first review by email

Line 111: How were any criteria modified, added or removed after the workshop? What was adapted and why?

=> The details of the criteria modified, added or removed are provided in the result section.

"Experts were overall satisfied by the checklist through the feedback that we collected as well as their interactive discussion in the workshop, but emphasized the importance of the investigators' training on safety reporting and GCP training of research managers to ensure engagement of the PI and study team towards a better compliance to current national and international GCP standards. Although the current Vietnamese laws do not regulate the recommended criteria covered in the checklist, experts felt compliance to these criteria should be closely monitored, as it would reveal active internal mechanisms deployed by local institutions to facilitate the implementation and the conduct of clinical trials per local and international standard guidelines (Figure 2)."

=> However, we agree that it should have been presented more clearly. To clarify the changes that occurred between the initial and final checklist we added the following

"Therefore, these two criteria, which were initially recommended criteria in the first draft version of the checklist developed by the MOH-ASTT team, were re-categorized into core criteria in the final checklist."

Line 138: Refers to Table 2, but this seems to not fit here

=> We corrected as this was "Figure 2"

Lines 140-168: Repeats what is in Figure 2. The reason for having this detail here is not clear.

=> We really hope that you will agree for us to keep it as the original version. We believe that it is clearer to have the details of the table outlined in the text. This presentation will help readers to

capture a view of each aspect, and criteria in each aspect. Besides, It will help the reader to understand all the abbreviations and their definition/meaning.

Lines 170-173: How was expert satisfaction determined?

Satisfaction was when all of the experts agreed about a criteria (its availability and classification (Core or Recommendation)? It was confirmed by their feedback sending to ASTT-MOH by email, and their interactive discussion in the workshop. We edited lines 204-205 to make it clearer

On the advice of experts and you suggesting there is a hierarchy in the criteria, that is, is investigator training more important than some other criteria?

=> No, there is not a complex hierarchy. The expert panel agreed with us that the hierarchy should contain only 2 levels, core and recommended, because when using in practice, we can only assess if a trial site is qualified or not. To be qualified, they have to meet all core criteria. Therefore, they should be equally important. If we used a more complicated hierarchy, its "regulatory" aspect would be less practical. The trial site would try to ask for compromises if they achieve some highest-level criteria and miss some "medium-high" level criteria, for instance.

Lines 247-250: The rationale for requiring a certificate to implement business activities is explained in the discussion but not before it is presented as a criteria. It may help to explain this prior.

=> Your comment is very reasonable since it is interesting criteria. We provided further explanation for this criteria in the discussion as most if not all of the studied sites did not comply to this criterion, requiring therefore a supportive discussion. Providing rationality of this criteria in the prior section would require the same for all the other criteria. It would massively increase the length of the manuscript and not necessarily benefit the readers. Therefore we have not introduced an explanation for this criterion earlier.

The discussion clearly explores the issues relating to recognition and reporting of adverse effects, and lack of SOP's and resources. Again, this is vital information for other resource-limited countries and the contribution of your work towards similar projects in these countries could be emphasised here.

=> Thank you for your kind suggestion. This section was added to answer the reviewer's comment: ""In this context, we believe that the adoption of a GCP inspection checklist in support of assessing clinical trial site is crucial for Vietnam, as well as other resource-limited countries, which are emerging in clinical trials. Thus, our GCP inspection checklist can serve as reference for resource-limited countries looking at strengthening clinical research capacity." We believe our work can benefit other countries and it is important to highlight this.

Figure 1 is not referred to in the text.

=> We referred it in the Line 89. But we move it into Line 89 (marked copy) for easier recognition according to your comment

VERSION 2 – REVIEW

REVIEWER	Hurtado-Chong, Anahi AO Foundation Office Dubendorf, AO ITC
REVIEW RETURNED	26-Jun-2021
GENERAL COMMENTS	The potential site selection bias was introduced in the limitations section but not in the discussion, where the original phrase regarding participation bias still remains as it was. Please make consistent. All the rest of my comments were properly addressed.
REVIEWER	Conroy, Tiffany

	Flinders University, College of Nursing and Health Sciences
REVIEW RETURNED	29-Jun-2021

GENERAL COMMENTS	thank you for your revisions and for addressing all of the reviewer suggestions.
--

VERSION 2 – AUTHOR RESPONSE

Reviewer: 1

Dr. Anahi Hurtado-Chong, AO Foundation Office Dubendorf

Comments to the Author:

The potential site selection bias was introduced in the limitations section but not in the discussion, where the original phrase regarding participation bias still remains as it was. Please make consistent. All the rest of my comments were properly addressed.

=> We revised these points according to your comment. We sincerely apologized for misunderstanding these points at the beginning and might not meet your requirements at the beginning. Please find our revision in the marked copy version of the main document